

# Association of multiple tumor markers with newly diagnosed gastric cancer patients: a retrospective study

Xiaoyang Li[1,*], Sifeng Li[2,*], Zhenqi Zhang[1] and Dandan Huang[1]

[1] Department of Medical Equipment, Sichuan Cancer Hospital, Chengdu, Sichuan, China
[2] Department of Medical, West China Tianfu Hospital, Sichuan University, Chengdu, Sichuan, China
* These authors contributed equally to this work.

## ABSTRACT

**Background:** The purpose of this paper was to explore the correlation between multiple tumor markers and newly diagnosed gastric cancer.

**Methods:** We selected 268 newly diagnosed patients with gastric cancer and 209 healthy subjects for correlation research. The detection of multiple tumor markers was based on protein chips and the results were statistically analyzed using SPSS.

**Results:** We concluded that gastric cancer was significantly related to gender, age, alpha fetoprotein (AFP), carcinoembryonic antigen (CEA), carbohydrate antigen 125 (CA125), carbohydrate antigen 199 (CA199), and carbohydrate antigen 242 (CA242) positive levels ($P < 0.001$). After CA199 and CA242 were stratified by gender, the male odds ratio (OR) was 30.400 and 31.242, respectively, while the female OR was 3.424. After CA125 was stratified by age in patients over 54 years old with gastric cancer, the risk of occurrence in the CA125-positive population was 16.673 times that of the CA125-negative patients. Among patients 54 years old and younger, being CA125-positive was not a risk factor for gastric cancer ($P = 0.082$). AFP, CEA, CA125, CA199, and CA242 positive levels during the M1 stage were statistically significant when compared with the M0 stage and control group ($P < 0.001$), but the AFP ($P = 0.045$) and CA125 ($P = 0.752$) positive levels were not statistically significant when compared with the M0 stage and control group. The combined detection sensitivity of multiple tumor markers was 44.78%.

**Conclusion:** Our research shows that gastric cancer is associated with age, gender, and the positive levels of AFP, CEA, CA125, CA199, and CA242. The positive levels of AFP and CA125 were related to the distant metastasis of gastric cancer. To a certain extent, the combined detection sensitivity can be used for the initial screening of gastric cancer.

Corresponding author
Dandan Huang, 375124932@qq.com

## INTRODUCTION

According to the 2019 State Council Development Research Center's report on cancer incentives and disease burden, gastric cancer ranks second in incidence of all cancers in China (*Wang Weijin, 2019*). Worldwide, the incidence and mortality rates of gastric cancer have been steadily declining over the last half century in most populations, although it was

responsible for over one million new cases in 2020 and an estimated 769,000 deaths (equating to one in every 13 cancer-related deaths). Gastric cancer ranks fifth in global incidence and fourth in mortality (*Sung et al., 2021*), and exceeds two-fifths of all new cancer cases diagnosed in China (*Wang Weijin, 2019*). Data has shown that the 5-year survival rate for patients whose gastric cancer was confined to the mucosal layer of the gastric wall was higher than that of advanced patients. Due to the lack of typical clinical symptoms in the early stages of gastric cancer, most patients are immediately diagnosed as advanced. The initial diagnosis rate is less than 10%. The 5-year survival rate is approximately 20% (*Wu et al., 2017*). Early detection and treatment are critical for improving the survival rate of patients. Because of the combination of early non-specific symptoms and objective factors such as a large population base and different medical standards in different regions, most gastric cancer patients are already in the middle and advanced stages when they are diagnosed, and treatment effects are poor.

Serum tumor markers are generally expressed at low levels in healthy patients or those with benign lesions. When the level is below the detection threshold, the test results are often negative. However, during cancer cell proliferation, the serum tumor marker level increases and can be used for the early screening of gastric cancer. It can also play a certain auxiliary role in the prognostic evaluation and diagnosis of gastric cancer (*Chen et al., 2012*; *Chen et al., 2017*; *He et al., 2013b*; *Lai et al., 2014*; *Lin et al., 2020*; *Wada et al., 2017*; *Yu, Zhang & Zhao, 2015*). The multi-tumor marker protein chip developed in China was used for the early diagnosis and monitoring of tumors by analyzing the content of 12 common tumor markers (carbohydrate antigen 199 (CA199), neuron-specific enolase (NSE), carcinoembryonic antigen (CEA), carbohydrate antigen 242 (CA242), ferritin (FER), β-HCG, alpha fetoprotein (AFP), prostate-specific antigen (PSA), free-PSA, carbohydrate antigen 125 (CA125), human growth hormone (HGH), and carbohydrate antigen 153 (CA153); *Sun et al., 2004*). In this study, we excluded the male and female-specific tumor markers (β-HCG, PSA, and free-PSA) and HGH, and only retained AFP, CEA, CA125, CA153, CA199, CA242, FER, and NSE.

Reference intervals (RIs) are defined as falling between the 2.5$^{th}$ and 97.5$^{th}$ percentiles of test result values obtained from a healthy population. Since only the high values are of clinical concern, we used a one-side RI, where the 95$^{th}$ percentile was used as the upper scale of the RI, following Clinical and Laboratory Standards Institute (CLSI) C28-A3 guidelines (*Horowitz, 2008*). It is common practice in clinic to cite the cut-off values provided by literature or a commercial kit, but this was not appropriate here because these cut-off values came from different laboratories, regions, populations, and instruments (*Bohn & Adeli, 2021*; *Jing et al., 2019*). Although the correlation between gastric cancer and eight tumor markers has been extensively studied, the data on positive rates of tumor markers and gastric cancer in Sichuan are very limited, and there has been no systematic evaluation of these eight markers on the same cohort using the Cochran–Mantel–Haenszel (CMH) test or logistic regression analysis.

The objectives of this paper were to study the relationship between newly diagnosed gastric cancer and tumor markers using a commercial multi-tumor marker detection kit; report the establishment of a reference interval for eight individual healthy biomarkers as

the threshold of our clinical study series using age-stratified, gender-stratified, and large cohort considerations; and further explore the possibility of early screening tumor markers in gastric cancer *via* logistic regression. Newly diagnosed gastric cancer hereon will be referred to gastric cancer.

## MATERIALS AND METHODS

### Materials

Our study protocol was approved by the Sichuan Cancer Hospital Ethics Committee (No. SCCHEC-02-2021-066). A total of 268 newly diagnosed gastric cancer patients (aged 60.91 ± 11.51) who were admitted to Sichuan Cancer Hospital between June 2018 and December 2019 were selected. The inclusion criteria was as follows: (1) Gastric cancer group: in line with the Guidelines for the Standardized Diagnosis and Treatment of Gastric Cancer (*China NHCotPsRo, 2013*) from the Chinese Medical Association, all diagnoses were performed by gastroscopy, CT, or B-ultrasound, and gastric cancer was confirmed by gastroscopy or postoperative pathological biopsy; no other primary cancer sites; no history of chemotherapy, radiotherapy, or immunotherapy before collecting serum samples; age > 18 years; M stage was collected according to the AJCC 7[th] edition. (2) Control group: 209 subjects who underwent physical examination in this hospital during the same period were selected as the control group (age 48.08 ± 12.66). Healthy physical examination: no serious heart, brain, liver, lung, kidney, or other primary diseases in the past, and relevant examinations were within the normal range; age > 18 years. Exclusion criteria: accompanied by major organ dysfunction, septic shock, hemorrhagic shock, myocardial infarction, benign tumors, any cancer or cancer history, recent hospitalizations or other diseases, pregnant or lactating. Signed patient informed consent was waived per committee approval, since patients could not be contacted and the research project did not involve private information or business interests.

The determination of the gastric cancer (considering the sample size met the minimum of 165, $\alpha = 0.01$, $\beta = 0.05$, $P_t = 0.2$, $Pc = 0.05$, $N_t:N_c = 1$) and control group (considering the sample size met the minimum of 120) sample sizes were based upon the Sample Size Calculations in Clinical Research, Third Edition (*Chow et al., 2017*) and the CLSI (*Horowitz, 2008*) guidelines.

## METHODS

### Equipment and reagents

The multi-tumor marker detection kit (Chaozhou Shukang Biotechnology Co., Ltd., Chaozhou, China), LU-07 biochip reader (Shanghai Mingyuan Shukang Biochip Co., Ltd., Shanghai, China), and biochip image analysis system software were obtained from Huzhou Shukang Biotechnology Co., Ltd.

We used a pyrogen-free and endotoxin test tube to draw 2 ml of fasting venous blood in the early morning the day after the patient's admission and excluded various factors that may have had an effect on the tumor markers. The serum samples were collected without hemolysis after centrifugation. After the serum was antiquated, it was stored in a

refrigerator at 4 °C, tested within 5 days, and equilibrated to room temperature before testing. We strictly followed the manufacturer's instructions for the biochip reader.

## Statistical analysis

The results were analyzed using SPSS26.0 statistical software (SPSS Inc., Chicago, IL, USA). The distribution of the data of the eight individual gastric cancer biomarkers was analyzed using the one-sample Kolmogorov-Smirnov and Shapiro-Wilk tests.
The analytical results determined whether the parametric or non-parametric statistical method would be used in the following analysis. After the normality test, the skewed distribution (non-normal distribution) was transformed using the logarithmic transformation.

To establish an RI for each of the eight gastric cancer biomarkers, we followed the CLSI C28-A3 guidelines. A 95th percentile was presented as the upper scale of the RI.
The normal cut-off values were as follows: AFP ≤ 3.69 ng/ml, NSE ≤ 4.33 ng/ml, FER ≤ 376.87 ng/ml, CEA ≤ 3.18 ng/ml, CA125 ≤ 20.46 U/ml, CA153 ≤ 24.46 U/ml, CA199 ≤ 24.23 U/ml, and CA242 ≤ 7.61 U/ml.

The chi-square test was utilized to analyze the clinical characteristics across the two groups. The age value was established using the decision tree method. Gender and age stratification statistics were tested using the CMH test. The area under curve (AUC) of the receiver operating characteristic (ROC) curve was used to evaluate the diagnostic value of the serum tumor markers. Logistic regression analysis was used to establish the diagnostic mathematical model. On the basis of this model, the prediction value was calculated, followed by ROC curve analysis. Bayes' theorem was used to evaluate the utility of the combined diagnostic indicators, Prevalence (age > 54) = 0.017858 (Cancer Today (iarc.fr)). $P < 0.010$ was considered statistically significant.

# RESULTS

## Descriptive statistics

After excluding participants who did not meet the baseline, a total of 268 gastric cancer patients and 209 healthy people were included in the analysis. Figure 1 is a participation flowchart of the gastric cancer and healthy controls included in this study. The main reasons for the exclusion of subjects included repeated testing, other treatments received, incomplete test information, and other diseases. Table 1 shows the normality test results of eight individual biomarkers. Our results indicate that datasets from all biomarkers had skewed distribution ($P < 0.001$). Most datasets that had been transformed using logarithmic transformation also had skewed distribution ($P < 0.001$). The data distribution for the tumor markers is shown in Table 2 and Fig. 2. The RIs of the eight tumor markers were calculated using the nonparametric method. For AFP, NSE, FER, CEA CA125, CA153, CA199, and CA242, the upper reference limits of the tumor markers were defined as the 95th percentile of the distribution. 90% confidence intervals of the reference limits were also calculated.

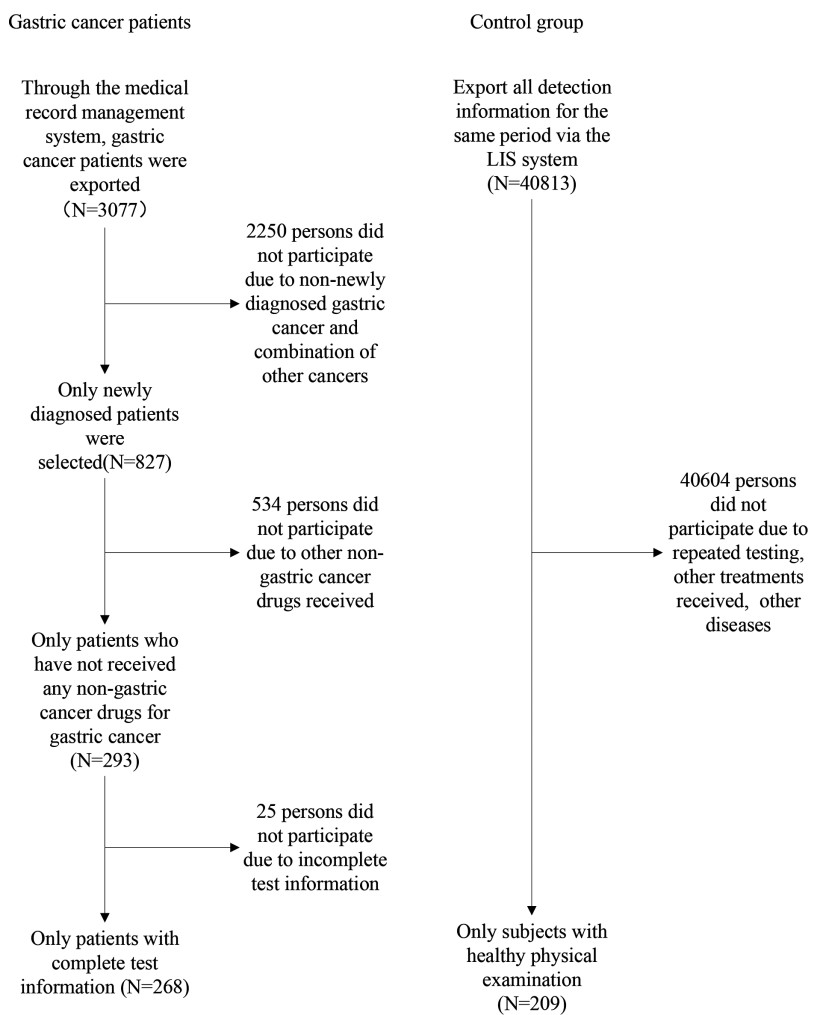

**Figure 1  Flowchart of participation in this study.**     

## Comparison of clinical characteristics between two groups

Tables 3 and 4 show that gastric cancer was significantly related to the patients' gender and age ($P < 0.001$). Males accounted for 73.9% of the gastric cancer group, males were 3.329 times more likely to have gastric cancer than females. The over-54 years old age group accounted for 70.89% of the gastric cancer patients, and they had a risk of gastric cancer that was 6.342 times that of the under-54 years patients.

Figure 2 shows that the positive levels of AFP, CEA, CA125, CA199, and CA242 were significantly correlated with gastric cancer ($P < 0.001$). To further explore the impact of gender and age on tumor markers, we conducted a group study based on gender and age.

## The relationship between gastric cancer and multiple tumor markers stratified by gender

Table 3 shows that the positive levels of AFP (OR = 3.803), CEA (OR = 6.633), CA125 (OR = 4.906), CA199, and CA242 were significantly correlated with gastric cancer ($P < 0.001$). After CA199 was stratified by gender, males had an OR of 30.400, the 95% CI

**Table 1 Normality test results of eight individual tumor markers.**

| Tumor markers | N | Kolmogorov–Smirnov | | Shapiro–Wilk | |
|---|---|---|---|---|---|
| | | Statistic | P | Statistic | P |
| AFP | Gastric cancer ($n = 268$) | 0.448 | 0.000 | 0.212 | 0.000 |
| | Control ($n = 209$) | 0.194 | 0.000 | 0.690 | 0.000 |
| NSE | Gastric cancer ($n = 268$) | 0.322 | 0.000 | 0.295 | 0.000 |
| | Control ($n = 209$) | 0.117 | 0.000 | 0.885 | 0.000 |
| FER | Gastric cancer ($n = 268$) | 0.174 | 0.000 | 0.820 | 0.000 |
| | Control ($n = 209$) | 0.102 | 0.000 | 0.934 | 0.000 |
| CEA | Gastric cancer ($n = 268$) | 0.363 | 0.000 | 0.455 | 0.000 |
| | Control ($n = 209$) | 0.185 | 0.000 | 0.662 | 0.000 |
| CA125 | Gastric cancer ($n = 268$) | 0.371 | 0.000 | 0.397 | 0.000 |
| | Control ($n = 209$) | 0.291 | 0.000 | 0.366 | 0.000 |
| CA153 | Gastric cancer ($n = 268$) | 0.314 | 0.000 | 0.341 | 0.000 |
| | Control ($n = 209$) | 0.192 | 0.000 | 0.714 | 0.000 |
| CA199 | Gastric cancer ($n = 268$) | 0.364 | 0.000 | 0.488 | 0.000 |
| | Control ($n = 209$) | 0.146 | 0.000 | 0.800 | 0.000 |
| CA242 | Gastric cancer ($n = 268$) | 0.370 | 0.000 | 0.441 | 0.000 |
| | Control ($n = 209$) | 0.157 | 0.000 | 0.700 | 0.000 |
| lg(AFP) | Gastric cancer ($n = 268$) | 0.101 | 0.000 | 0.877 | 0.000 |
| | Control ($n = 209$) | 0.117 | 0.000 | 0.945 | 0.000 |
| lg(NSE) | Gastric cancer ($n = 268$) | 0.145 | 0.000 | 0.815 | 0.000 |
| | Control ($n = 209$) | 0.050 | 0.200 | 0.987 | 0.049 |
| lg(FER) | Gastric cancer ($n = 268$) | 0.086 | 0.000 | 0.959 | 0.000 |
| | Control ($n = 209$) | 0.115 | 0.000 | 0.909 | 0.000 |
| lg(CEA) | Gastric cancer ($n = 268$) | 0.151 | 0.000 | 0.886 | 0.000 |
| | Control ($n = 209$) | 0.060 | 0.065 | 0.988 | 0.078 |
| lg(CA125) | Gastric cancer ($n = 268$) | 0.185 | 0.000 | 0.845 | 0.000 |
| | Control ($n = 209$) | 0.091 | 0.000 | 0.911 | 0.000 |
| lg(CA153) | Gastric cancer ($n = 268$) | 0.117 | 0.000 | 0.930 | 0.000 |
| | Control ($n = 209$) | 0.096 | 0.000 | 0.978 | 0.003 |
| lg(CA199) | Gastric cancer ($n = 268$) | 0.149 | 0.000 | 0.898 | 0.000 |
| | Control ($n = 209$) | 0.039 | 0.200 | 0.992 | 0.279 |
| lg(CA242) | Gastric cancer ($n = 268$) | 0.186 | 0.000 | 0.830 | 0.000 |
| | Control ($n = 209$) | 0.047 | 0.200 | 0.984 | 0.015 |

Note:
All biomarkers had skewed distribution ($p < 0.001$). Only CA199 and CEA in the control group could be transformed into normal distribution by logarithmic transformation.

was [4.127–223.928], $P < 0.001$, and the risk of CA199-positive patients being diagnosed with gastric cancer was 30.4 times greater than CA199-negative patients. In females, being CA199-positive was also a risk factor for gastric cancer. Their OR was 3.424, the 95% CI was [1.420–8.257], $P = 0.004$, and the risk of CA199-positive patients being diagnosed with gastric cancer was 3.424 times greater than CA199-negative patients. After CA242 was stratified by gender, males had an OR of 31.242, the 95% CI was [4.243–230.043],

**Table 2 Distribution, RIs of tumor marker levels, and 90% confidence intervals.**

| Tumor markers | Gastric cancer (n = 268) | Control (n = 209) | | |
|---|---|---|---|---|
| | M (IQR) | M (IOR) | Reference intervals | 90% CI |
| AFP (ng/ml) | 1.36 (2.05) | 1.26 (0.94) | 0–3.69 | [3.02–5.49] |
| NSE (ng/ml) | 2.59 (1.09) | 2.55 (1.21) | 0–4.33 | [4.17–4.96] |
| FER (ng/ml) | 76.91 (160.11) | 138.03 (202.03) | 0–376.87 | [365.96–424.65] |
| CEA (ng/ml) | 1.47 (2.39) | 1.07 (0.94) | 0–3.18 | [2.77–3.82] |
| CA125 (U/ml) | 7.31 (10.14) | 8.55 (5.65) | 0–20.46 | [16.82–32.90] |
| CA153 (U/ml) | 4.54 (3.75) | 7.01 (7.62) | 0–24.46 | [21.27–27.91] |
| CA199 (U/ml) | 10.35 (18.61) | 7.47 (7.36) | 0–24.23 | [19.45–29.37] |
| CA242 (U/ml) | 3.28 (4.79) | 3.04 (2.01) | 0–7.61 | [6.70–8.95] |

$P < 0.001$, and the risk of CA242-positive patients being diagnosed with gastric cancer was 31.242 times greater than CA242-negative patients. In females, being CA242-positive was also a risk factor for gastric cancer. Their OR was 3.424, the 95% CI was [1.420–8.257], $P = 0.004$, and the risk of CA242-positive patients being diagnosed with gastric cancer was 3.424 times greater than CA242-negative patients.

## The relationship between gastric cancer and multiple tumor markers stratified by age

Table 4 shows that the positive levels of AFP (OR = 3.803), CEA (OR = 6.633), CA125, CA199 (OR = 6.234), and CA242 (OR = 6.372) was significantly related to gastric cancer ($P < 0.001$), After CA125 was stratified by age, patients older than 54 years had an OR of 16.673, the 95% CI was [2.243–123.91], $P < 0.001$, and the risk of CA125-positive patients being diagnosed with gastric cancer was 16.673 times that of CA125-negative patients. In patients 54 years-old or younger, being CA125-positive was not a risk factor for the disease. Their OR was 2.32, the 95% CI was [0.901–5.974], and $P = 0.082$.

## The relationship between gastric cancer distant metastasis and tumor markers

We further studied the relationship between gastric cancer distant metastasis and tumor markers. Table 5 and Fig. 3 show that with the progression of gastric cancer, the proportion of positive tumor markers increased, and the AFP, CEA, CA125, CA199, and CA242 positive levels in the M1 stage were statistically significant compared with those in the M0 stage and control group ($P < 0.001$), although the AFP ($P = 0.045$) and CA125 ($P = 0.752$) positive levels were not statistically significant when compared with the M0 stage and control group.

## Sensitivity, specificity, and accuracy of single and combined tumor markers with gender and age

The results in Table 6 show that the sensitivity of AFP, CEA, CA125, CA199, and CA242 to detect gastric cancer was 16.04%, 25.00%, 19.78%, 23.88%, and 24.25%, respectively, and

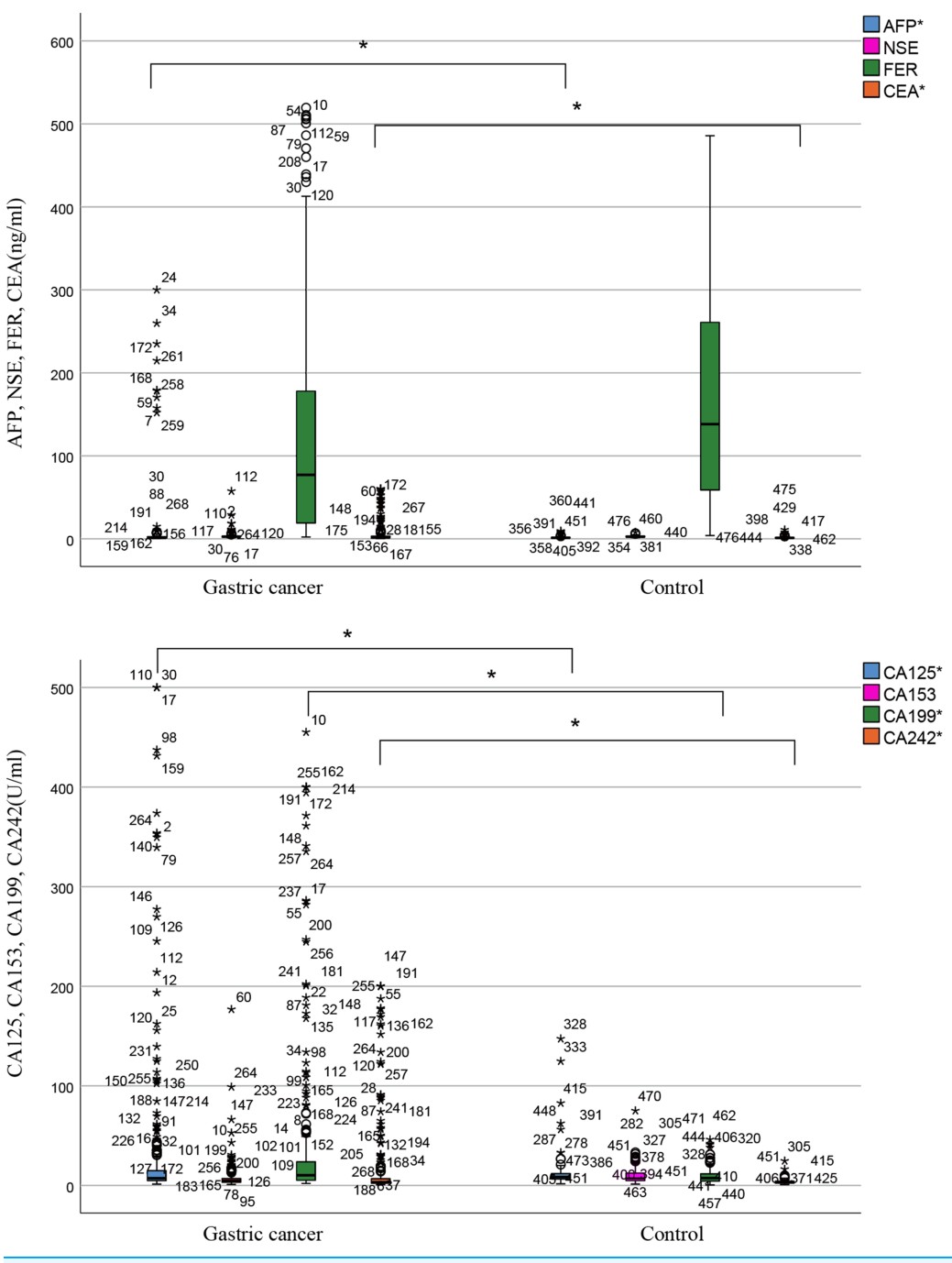

**Figure 2 Distribution of tumor marker levels with gastric cancer and control groups (box and whisker plot).** The distribution of AFP, NSE, FER, CEA, CA125, CA153, CA199, and CA242 levels in gastric cancer and control groups; *P < 0.001.

the sensitivity of combined detection was 44.78%, with a specificity of 95.22% and an accuracy of 66.88%. In the M0 stage, the sensitivity of AFP, CEA, CA125, CA199, and CA242 to detect gastric cancer was 9.95%, 16.92%, 5.47%, 14.43%, 14.93%, respectively, and the sensitivity of combined detection was 35.82%. In the M1 stage, the sensitivity of AFP, CEA, CA125, CA199, and CA242 to detect gastric cancer was 34.33%, 49.25%,

**Table 3 The relationship between gastric cancer and multiple tumor markers stratified by gender.**

| Tumor markers | Gender | Gastric cancer (n = 268) | Control (n = 209) | χ² | OR 95% CI | P |
|---|---|---|---|---|---|---|
| / | Female | 70 | 113 | | | |
| / | Male | 198 | 96 | 38.787 | 3.329 [2.265–4.893] | <0.001 |
| AFP- | Female | 60 | 105 | | | |
| | Male | 165 | 94 | | | |
| AFP+ | Female | 10 | 8 | | | |
| | Male | 33 | 2 | 15.043 | 3.803 [1.862–7.767] | <0.001 |
| NSE− | Female | 62 | 110 | | | |
| | Male | 179 | 89 | | | |
| NSE+ | Female | 8 | 3 | 4.592 | 2.229 [1.054–4.717] | 0.038 |
| | Male | 19 | 7 | | | |
| FER− | Female | 67 | 113 | | | |
| | Male | 185 | 86 | | | |
| FER+ | Female | 3 | 0 | 4.924 | / | 0.054 |
| | Male | 13 | 10 | 1.330 | 0.604 [0.255–1.433] | 0.249 |
| CEA− | Female | 55 | 109 | | | |
| | Male | 146 | 90 | | | |
| CEA+ | Female | 15 | 4 | 35.449 | 6.633 [3.318–13.261] | <0.001 |
| | Male | 52 | 6 | | | |
| CA125− | Female | 54 | 104 | | | |
| | Male | 161 | 95 | | | |
| CA125+ | Female | 16 | 9 | 23.022 | 4.906 [2.429–9.905] | <0.001 |
| | Male | 37 | 1 | | | |
| CA153− | Female | 66 | 109 | | | |
| | Male | 191 | 90 | | | |
| CA153+ | Female | 4 | 4 | 0.129 | 0.852 [0.355–2.046] | 0.719 |
| | Male | 7 | 6 | | | |
| CA199− | Female | 54 | 104 | | | |
| | Male | 150 | 95 | | | |
| CA199+ | Female | 16 | 9 | 8.128 | 3.424 [1.420–8.257] | 0.004 |
| | Male | 48 | 1 | 25.057 | 30.400 [4.127–223.928] | <0.001 |
| CA242− | Female | 54 | 104 | | | |
| | Male | 149 | 95 | | | |
| CA242+ | Female | 16 | 9 | 8.128 | 3.424 [1.644–4.446] | 0.004 |
| | Male | 49 | 1 | 25.741 | 31.242 [4.243–230.043] | <0.001 |

**Note:**
OR, odds ratio; CI, confidence interval.

62.69%, 52.24%, 52.24%, respectively, and the sensitivity of combined detection was 71.64%. Bayes' theorem was also used to evaluate the utility of the combined diagnostic indicators based on the prevalence of gastric cancer. The sensitivity was 14.55% the specificity was 98.96% (Table 6).

**Table 4 The relationship between gastric cancer and multiple tumor markers stratified by age.**

| Variable | Age | Gastric cancer (n = 268) | Control (n = 209) | $\chi^2$ | OR 95% CI | P |
|---|---|---|---|---|---|---|
| / | ≤54 | 78 | 151 | | | |
| / | >54 | 190 | 58 | 87.571 | 6.342 [4.245–9.474] | <0.001 |
| AFP− | ≤54 | 69 | 144 | | | |
| | >54 | 156 | 55 | | | |
| AFP+ | ≤54 | 9 | 7 | 15.074 | 3.803 [1.862–7.767] | <0.001 |
| | >54 | 34 | 6 | | | |
| NSE− | ≤54 | 73 | 141 | | | |
| | >54 | 168 | 58 | | | |
| NSE+ | ≤54 | 5 | 10 | 4.592 | 2.580 [1.050–6.336] | 0.032 |
| | >54 | 22 | 0 | | | |
| FER− | ≤54 | 76 | 141 | | | |
| | >54 | 176 | 58 | | | |
| FER+ | ≤54 | 2 | 10 | 1.706 | 0.371 [0.079–1.737] | 0.229 |
| | >54 | 14 | 0 | 4.529 | 1.330 [1.235–1.431] | 0.045 |
| CEA− | ≤54 | 67 | 146 | | | |
| | >54 | 134 | 53 | | | |
| CEA+ | ≤54 | 11 | 5 | 35.449 | 6.633 [3.318–13.261] | <0.001 |
| | >54 | 56 | 5 | | | |
| CA125− | ≤54 | 68 | 142 | | | |
| | >54 | 147 | 57 | | | |
| CA125+ | ≤54 | 10 | 9 | 3.181 | 2.32 [0.901–5.974] | 0.082 |
| | >54 | 43 | 1 | 13.309 | 16.673 [2.243–123.91] | <0.001 |
| CA153− | ≤54 | 75 | 142 | | | |
| | >54 | 182 | 57 | | | |
| CA153+ | ≤54 | 3 | 9 | 0.129 | 0.852 [0.355–2.046] | 0.719 |
| | >54 | 8 | 1 | | | |
| CA199− | ≤54 | 66 | 144 | | | |
| | >54 | 138 | 55 | | | |
| CA199+ | ≤54 | 12 | 7 | 32.670 | 6.243 [3.117–12.503] | <0.001 |
| | >54 | 52 | 3 | | | |
| CA242− | ≤54 | 66 | 144 | | | |
| | >54 | 137 | 55 | | | |
| CA242+ | ≤54 | 12 | 7 | 33.589 | 6.372 [3.184–12.753] | <0.001 |
| | >54 | 53 | 3 | | | |

**Note:**
OR, odds ratio; CI, confidence interval.

# DISCUSSION

The prevalence of gastric cancer increases with age, and males are at a greater risk than females (*Joshi & Badgwell, 2021*; *Sung et al., 2021*; *Yang, Zheng & Zhang, 2019*; *Zuo et al., 2017*). In this study, the number of male patients was much more than that of female patients, and the male-female sex ratio was 3.36:1. The difference between patients 54 years

**Table 5 Difference of tumor markers between M stages.**

| Variable | AFP+ | AFP− | CEA+ | CEA− | CA125+ | CA125− | CA199+ | CA199− | CA242+ | CA242− |
|---|---|---|---|---|---|---|---|---|---|---|
| Control | 10 | 199 | 10 | 199 | 10 | 199 | 10 | 199 | 10 | 199 |
| M0(201) | 20 | 181 | 34 | 167 | 11 | 190 | 29 | 172 | 30 | 171 |
| M1(67) | 23 | 44 | 33 | 34 | 42 | 25 | 35 | 32 | 35 | 32 |
| $\chi^2 1$ | 4.031 | | 15.739 | | 0.100 | | 11.069 | | 11.967 | |
| P1 | 0.045* | | <0.001 | | 0.752* | | 0.001 | | 0.001 | |
| $\chi^2 2$ | 42.067 | | 76.282 | | 111.241 | | 83.725 | | 83.725 | |
| P2 | <0.001 | | <0.001 | | <0.001 | | <0.001 | | <0.001 | |
| $\chi^2 3$ | 22.169 | | 28.027 | | 103.680 | | 39.521 | | 38.083 | |
| P3 | <0.001 | | <0.001 | | <0.001 | | <0.001 | | <0.001 | |

Notes:
$\chi^2 1$,P1 M0 *vs*. control, $\chi^2 2$,P2 M1 *vs*. control, $\chi^2 3$,P3 M0 *vs*. M1.
* AFP ($P = 0.045$) and CA125 ($P = 0.752$) positive levels were not statistically significant when comparing the M0 stage and control group.

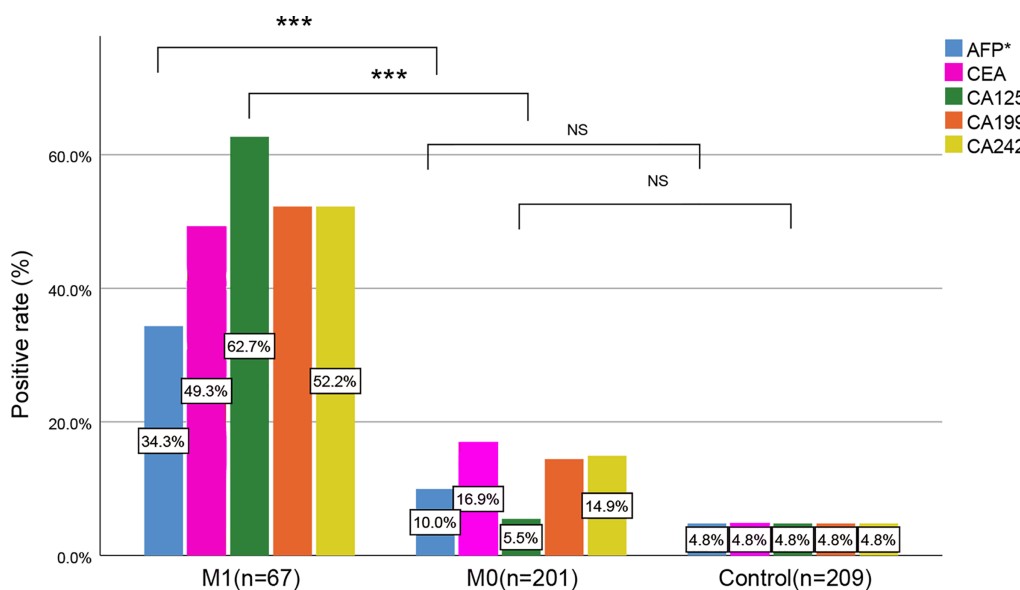

**Figure 3 Comparison of tumor markers among three groups.** *AFP, CA125 in the M1 stage were statistically significant when compared with the M0 stage and the control group, but AFP and CA125 positive levels were not statistically significant when comparing the M0 stage and control group; NS, no significance; ***$P < 0.001$.

and younger and patients older than 54 years was statistically significant. The risk of gastric cancer for patients older than 54 years was 6.342 times that of patients 54 years old and younger.

AFP is widely used in the diagnosis of gastric cancer (*He et al., 2013a*; *Matsuoka & Yashiro, 2018*). The results of our study showed that the sensitivity of AFP in the gastric cancer group was only 16.04%, and AFP positive levels in the M1 stage were statistically significant compared with those in the M0 stage and control group ($P < 0.001$), although the positive of AFP ($P = 0.045$) was not statistically significant when comparing the M0 stage with the control group. This suggested that AFP may be related to the distant

**Table 6 Sensitivity, specificity, accuracy, combination, and evaluation of tumor markers.**

| Tumor marker | AUC (95% CI) | Cut-off value | Sensitivity (%) | | | Specificity (%) | Accuracy (%) |
|---|---|---|---|---|---|---|---|
| | | | M0+M1 | M0 | M1 | | |
| AFP | 0.539 [0.487–0.590] | 3.69 | 16.04 (43/268) | 9.95 (20/201) | 34.33 (23/67) | 95.22 (199/209) | 50.73 (242/477) |
| CEA | 0.641 [0.592–0.690] | 3.18 | 25.00 (67/268) | 16.92 (34/201) | 49.25 (33/67) | 95.22 (199/209) | 55.77 (266/477) |
| CA125 | 0.451 [0.400–0.503] | 20.46 | 19.78 (53/268) | 5.47 (11/201) | 62.69 (42/67) | 95.22 (199/209) | 52.83 (252/477) |
| CA199 | 0.607 [0.557–0.657] | 24.23 | 23.88 (64/268) | 14.43 (29/201) | 52.24 (35/67) | 95.22 (199/209) | 55.14 (263/477) |
| CA242 | 0.557 [0.506–0.609] | 7.61 | 24.25 (65/268) | 14.93 (30/201) | 52.24 (35/67) | 95.22 (199/209) | 55.35 (264/477) |
| Combination[1] | 0.845 [0.811–0.879] | 0.8057495 | 44.78 (120/268) | 35.82 (72/201) | 71.64 (48/67) | 95.22 (199/209) | 66.88 (319/477) |
| Evaluation[2] | / | / | 14.55[a] | 11.99[a] | 21.42[a] | 98.96[b] | / |

Notes:
[1] Logit (P) = −1.19 * gender + 0.08 * age + 0.013 * AFP + 0.091 * CEA + 0.008 * CA125 + 0.036 * CA199 − 0.039 * CA242 − 4.332.
[2] Prevalence (age > 54) = 0.017858.
[a] Sensitivity * Prevalence/(Sensitivity * Prevalence + (1−Specificity) * (1−Prevalence)).
[b] Specificity * (1−Prevalence)/(Specificity * (1−Prevalence) + (1−Sensitivity) * Prevalence).

metastasis of gastric cancer. Previous research have connected being AFP-positive with liver metastasis (*Guan et al., 2018*; *Liu et al., 2010*).

CEA is synthesized in small amounts in the gastrointestinal tract of adults and is excreted through the gastrointestinal tract without entering the blood system. When gastrointestinal tumors occur, the expression of CEA in serum can significantly increase (*Lai et al., 2002*; *Matsuoka & Yashiro, 2018*; *Ucar et al., 2008*). *Fangning et al. (2020)* tested 3,807 gastric cancer patients and found 756 patients with positive CEA levels, with a sensitivity of 19.9% in patients with gastric cancer. This study showed that the sensitivity of serum CEA in patients with gastric cancer was 25.00%, which directly confirmed that CEA was highly expressed in patients with gastric cancer.

CA125 is generally used in the diagnosis and prognosis of ovarian cancer. Related studies have shown that the sensitivity of CA125 to gastric cancer is 34.3% (*Namikawa et al., 2018*), and the results of this study showed that the sensitivity was 19.78%. CA125 positive levels in the M1 stage were statistically significant compared with those in the M0 stage and the control group ($P < 0.001$), but CA125 ($P = 0.752$) was not statistically significant when the M0 stage was compared to the control group. This conclusion suggested that CA125 might be related to the distant metastasis of gastric cancer. CA125 positive levels have shown significant elevations in the presence of peritoneal carcinomatozis (*Polat et al., 2014*), and *Namikawa et al. (2018)* found that CA125 is a useful prognostic biomarker in patients with unresectable advanced or recurrent gastric cancer. Multiple studies have shown that CA125 is related to peritoneal metastasis (*Lai et al., 2014*; *Luo et al., 2016*; *Peng et al., 2014*; *Zhao et al., 2017*). Our study validates these results.

CA199 is reported to have the highest sensitivity in the diagnosis of gastrointestinal tumors (*Wang et al., 2014*). This study found that the sensitivity of CA199 was only 23.88%, which was not much different from the sensitivity (19.0%) found by *Fangning et al. (2020)*.

CA242 is a mucin-like glycoprotein. *Zhao et al. (2016)* and others found that the sensitivity of serum CA242 when used to detect gastric cancer was as high as 25~60%. The results of this study showed that the sensitivity of serum CA242 in patients with gastric cancer was only 24.25%. Due to different tumor stages, regions, genders, and study subjects, the diagnostic sensitivity of the research results was slightly different. At the same time, the results of this study could provide a quantitative reference for the clinical diagnosis of gastric cancer. Additionally, the CMH test in this study showed that although CA199 and CA242 were risk factors for gastric cancer, the risk factors were different due to the influence of gender. Males had a greater risk value. The results of age stratification showed that for people older than 54 years old, being CA125-positive was a risk factor, but for people 54 years old and younger, it was not a risk factor.

Thus far, no ideal tumor markers with 100% sensitivity and 100% specificity have been found in gastric cancer detection. Our results showed that the sensitivity of the single detection of tumor markers was low, ranging from 16.04% to 25.00%. Because the detection capacity of individual tumor markers was very limited (AUC, 0.451–0.641), the combined detection of multiple indicators could be used to make up for some of these shortcomings. *Yang et al. (2008)* found that the overall sensitivity of multi-tumor markers was only 35.9%. In this study, the comprehensive sensitivity for detecting gastric cancer was 44.78% (AUC = 0.845).

Our research and previous studies (*Feng et al., 2017*; *He et al., 2013a*; *Joshi & Badgwell, 2021*) showed that gastric cancer is related to age, gender, and individual tumor markers, and the combined detection of multi-index tumor markers can be used for early screening of gastric cancer to a certain extent. However, there are also some limitations. According to our clinical test results of tumor markers, the data was seriously skewed, similarly to *Zhou, Zhao & Shen (2015)*, and cannot be quantitatively analyzed through data transformation or processing (Table 1). In this study, we reconfirmed that the serum tumor markers were not normally distributed. Our analysis and statistical methods were improved, our statistical method used the CMH test and logistic regression, and we took into full account the impact of gender, age, and tumor markers. Using combined diagnostics based on the prevalence of gastric cancer in China was also evaluated using Bayes' theorem. This is one point that distinguishes this study from others. Additionally, the sample size was larger than in other studies, and there were more types of tumor markers detected. We also used a biochip detection method, which guarantees quality control.

However, there were a few limitations to our study. First, this was an analysis that used commercial kits and relatively limited tumor markers. Therefore, some newly discovered tumor markers could not be used in time for testing. Second, some patients were diagnosed as advanced, but the gastric cancer group was not subdivided by tumor stage, nor did we study the relationship between tumor location and tumor markers. Third, multi-tumor marker testing is not included in the scope of physical examinations for healthy people, so the sample size of the control group was small.

## CONCLUSION

In conclusion, gastric cancer is associated with age, gender, and the positive levels of AFP, CEA, CA125, CA199, and CA242. The positive levels of AFP and CA125 are related to the distant metastasis of gastric cancer. Combined detection based on logistic regression analysis can be used for initial screening of gastric cancer to a certain extent.

To improve the early diagnosis of gastric cancer and to reduce the missed diagnosis rate, higher-risk populations should first be identified through tumor marker detection, then imaging, gastroscopy, and colonoscopy. This will be more conducive to systematic digestive malignant tumor detection, and has easily obtainable materials, a convenient operation, and low cost, making it feasible in large-scale gastric cancer screenings. Multi-tumor marker detection can screen a variety of common cancers, and early intervention and treatment are key to reducing the medical burden of the country and individuals. Joint testing for multiple tumor markers should be advocated.

### Funding

This work was supported by the Sichuan Cancer Hospital. The funders had no role in study design, data collection and analysis, decision to publish, or preparation of the manuscript.

### Grant Disclosures

The following grant information was disclosed by the authors:
Sichuan Cancer Hospital.

### Competing Interests

The authors declare that they have no competing interests.

### Author Contributions

- Xiaoyang Li conceived and designed the experiments, performed the experiments, analyzed the data, prepared figures and/or tables, authored or reviewed drafts of the paper, and approved the final draft.
- Sifeng Li conceived and designed the experiments, performed the experiments, analyzed the data, prepared figures and/or tables, authored or reviewed drafts of the paper, and approved the final draft.
- Zhenqi Zhang conceived and designed the experiments, prepared figures and/or tables, and approved the final draft.
- Dandan Huang analyzed the data, authored or reviewed drafts of the paper, and approved the final draft.

## Human Ethics

The following information was supplied relating to ethical approvals (*i.e.*, approving body and any reference numbers):

Sichuan Cancer Hospital Ethics Committee approval to carry out the study within its facilities (No. SCCHEC-02-2021-066).

## Data Availability

The raw data is available in the Supplemental File.

## Supplemental Information

Supplemental information for this article can be found online at http://dx.doi.org/10.7717/peerj.13488#supplemental-information.

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
