# Peer review of "Association of multiple tumor markers with newly diagnosed gastric cancer patients: a retrospective study"

_PeerJ, doi:10.7717/peerj.13488_

## Round 0.1 · original submission · Major Revisions

When assessing your paper, the reviewers identified major issues that must be addressed for the manuscript to be suitable for publication in this journal.

Reviewer 1 ·

Basic reporting

The language of this manuscript is substandard. Grammatical errors severely compromised readability and lack of coherence made the manuscript hard to follow.

Figure and table legends failed to provide enough information for readers to interpret what they see. Figures and tables are poorly designed. Y-axis in figures comes with no labels and no indication of significance is shown in histograms.

Unsubstantiated and odd statements scatter in the manuscript. Examples are lines 62-65, “Most literature is used to determine the appropriate detection threshold through the ROC curve, although the results can significantly improve the sensitivity to detect gastric cancer, but the screening efficiency is reduced, healthy people have a higher chance of testing false positive”; lines 117-119 “The data distribution of tumor markers was shown in Table 1 and Fig 2. And was seriously skewed and cannot be quantitatively analyzed through data transformation or processing”; and similarly, lines 249-250 "According to the clinical test results of tumor markers, the data are seriously skewed and cannot be quantitatively analyzed through data transformation or processing.

The manuscript claims to “report the establishment of a reference interval for eight individual healthy people biomarkers” (lines 65-70). However, there is no information about how these reference intervals are established nor what the intervals are.

Experimental design

The authors have not clearly described their experimental design. It seems that this is a stratified case-cohort study, but it is confusing to read in the manuscript that this is a “phase I” clinical study (line 69).

Figure 1 is supposed to be a summary of subject selection. In the flowchart, the authors presented that after “screening outpatients who have not received any treatment for gastric cancer (N=293)” from the 827 patients who were newly diagnosed with gastric cancer, “534 persons did not participate”. It is very confusing because this implies that 534 newly diagnosed gastric cancer patients have already received treatment for gastric cancer. It doesn't make logical sense.

The authors provided little baseline information about the control group. Does the control group consists of only healthy individuals, or does it also includes individuals with no gastric cancer but have similar symptoms? Neglecting to include these patients might falsely inflate the performance of the test.

The authors have not explained how they determined sample size to reach adequate statistical power.

To establish a diagnostic tool with a “combination” of serum markers, the authors shall generate a regression model with the markers of interest, construct a ROC curve and calculate the AUC based on the predicted probabilities. Simply adding numbers up as shown in table 5 does not yield a clinically meaningful combination.

Finally, If such a diagnostic tool involving a "combination" of serum markers exists, the authors neglected to evaluate the utility of this tool based on the prevalence of gastric cancer.

Validity of the findings

The correlation between gastric cancer and multiple tumor markers in the manuscript has been extensively studied. I could not see the rationale for this replication here.

Feng, Fan, et al. "Diagnostic and prognostic value of CEA, CA19–9, AFP and CA125 for early gastric cancer." BMC cancer 17.1 (2017): 1-6.
Yin, Li-Kui, Xue-Qing Sun, and Dong-Zhen Mou. "Value of combined detection of serum CEA, CA72-4, CA19-9 and TSGF in the diagnosis of gastric cancer." Asian Pacific Journal of Cancer Prevention 16.9 (2015): 3867-3870.
He, Chao-Zhu, et al. "Combined use of AFP, CEA, CA125 and CAl9-9 improves the sensitivity for the diagnosis of gastric cancer." BMC gastroenterology 13.1 (2013): 1-5.

Additional comments

None

Reviewer 2 ·

Basic reporting

Li et.al., did a good job providing background information and as well as context for the study but the background needs to be further improved to bring more clarity for the readers. Few instances below:

1. Li et.al., needs to provide sufficient context and rationale as to why these biomarkers were included (and if some were considered and excluded) for gastric cancer study.
2. Li et.al., should provide clarity on what exactly they mean by reference value and reference interval in the introduction section.
3. Li et.al., should further improve the English language to make it amenable for international audience to understand the work (e.g., Line 59 "It is common practice in clinical practice" should be replaced to "It is common practice in clinic"). There are random capitalization throughout the document which needs to be corrected (e.g., Capital M in "Most" on line 62)

Experimental design

Li et.al., did a good job with the data analysis and tried it answer the key question if there is a correlation between multiple tumor biomarkers with newly diagnosed gastric cancer and concluded that gastric cancer is associated with AFP, CEA, CA125, CA199 and CA242. However more context needs to be provided to make the case

1. Li.et.al., needs to explain in the method what was the cut-off value used to define a positive tumor biomarker
2. Li.et.al., needs to explain in the method when was the sample collected preoperative or postoperative and if it has any impact on the outcome

Validity of the findings

Li et. al., have shown through statistically analysis that the serum biomarkers are correlated to age, gender of newly diagnosed gastric cancer. Some improvement in text might be helpful for more clarity. For example:

1. Gastric cancer is used repeatedly in the text instead of newly diagnosed gastric cancer. Either such statements should be clarified early on (e.g., Newly diagnosed gastric cancer hereon will be referred to gastric cancer)
2. Flowchart needs more clarity. Readers may find the process chart confusing. It would be helpful to clarify it a little bit (e.g., 2250 persons didn't participate due to XXXX instead of "2250 persons didn't participate). Similarly "Only newly diagnosed patients were selected" instead of screening out newly diagnosed patients with gastric cancer (n=827)
3. Y axis on Figure 2 and Figure 3 needs to be specified.

---

## Round 0.2 · accepted · Accept

Thank you for submitting your manuscript.

Reviewer 1 ·

Basic reporting

no comment

Experimental design

no comment

Validity of the findings

no comment

Additional comments

The authors have addressed all my concerns. I would recommend it for publication on PeerJ.

Reviewer 2 ·

Basic reporting

No comment

Experimental design

No Comment

Validity of the findings

No Comment

Additional comments

The authors have addressed all the comments appropriately.